# Comparison of Apparent Diffusion Coefficient Values on Diffusion-Weighted MRI for Differentiating Hepatocellular Carcinoma and Intrahepatic Cholangiocarcinoma

**DOI:** 10.3390/diagnostics15151861

**Published:** 2025-07-24

**Authors:** Katrīna Marija Konošenoka, Nauris Zdanovskis, Aina Kratovska, Artūrs Šilovs, Veronika Zaiceva

**Affiliations:** 1Department of Radiology, Riga Stradins University, Dzirciema Street, 16, LV-1007 Riga, Latvia; 036336@rsu.edu.lv (K.M.K.); aina.kratovska@aslimnica.lv (A.K.); arturs.silovs@rsu.lv (A.Š.); 2Department of Interventional Radiology, Riga East University Hospital, Hippocrates St., 2, LV-1038 Riga, Latvia; veronika.zaiceva@aslimnica.lv

**Keywords:** hepatocellular carcinoma, cholangiocarcinoma, intrahepatic, magnetic resonance imaging, apparent diffusion coefficient, tumor grading, immunohistochemistry, biopsy, liver neoplasms

## Abstract

**Background and Objectives**: Accurate noninvasive differentiation between hepatocellular carcinoma (HCC) and intrahepatic cholangiocarcinoma (ICC) remains a clinical challenge. This study aimed to assess the dignostic performance of apparent diffusion coefficient (ADC) values from diffusion-weighted MRI in distinguishing between HCC and ICC, with histological confirmation as the gold standard. **Materials and Methods**: A retrospective analysis was performed on 61 patients (41 HCC, 20 ICC) who underwent liver MRI and percutaneous biopsy between 2019 and 2024. ADC values were measured from diffusion-weighted sequences (b-values of 0, 500, and 1000 s/mm^2^), and regions of interest were placed over solid tumor areas. Statistical analyses included *t*-tests, one-way ANOVA, and ROC curve analysis. **Results**: Mean ADC values did not differ significantly between HCC (1.09 ± 0.19 × 10^−3^ mm^2^/s) and ICC (1.08 ± 0.11 × 10^−3^ mm^2^/s). ROC analysis showed poor discriminative ability (AUC = 0.520; *p* = 0.806). In HCC, ADC values decreased with lower differentiation grades (*p* = 0.008, η^2^ = 0.224). No significant trend was observed in ICC (*p* = 0.410, η^2^ = 0.100). Immunohistochemical markers such as CK-7, Glypican 3, and TTF-1 showed significant diagnostic value between tumor subtypes. **Conclusions**: ADC values have limited utility for distinguishing HCC from ICC but may aid in HCC grading. Immunohistochemistry remains essential for accurate diagnosis, especially in poorly differentiated tumors. Further studies with larger cohorts are recommended to improve noninvasive diagnostic protocols.

## 1. Introduction

Primary liver cancer is a significant health concern globally. It is the sixth leading cause of cancer-related mortality in Europe and the third most common worldwide. The global burden of liver cancer continues to rise, and projections suggest that, without improvements in prevention and control strategies, the number of liver cancer cases in Europe will increase by approximately 22% by the year 2040. This trend is largely attributed to population aging [1]. Hepatocellular carcinoma (HCC) and intrahepatic cholangiocarcinoma (ICC) are the two major histological subtypes of primary liver cancer. Among all primary liver malignancies, HCC accounts for approximately 80% of cases, while ICC represents around 15%. The remaining 5% consists of rarer subtypes [2]. Diagnosis is further complicated by combined hepatocellular–cholangiocarcinoma (cHCC-CC) variants, which exhibit overlapping features of both tumor types [3,4].

Hepatocellular carcinoma (HCC) is the most common primary liver malignancy. It originates from hepatocytes and retains morphological similarities to normal liver cells. Tumor grading is based on the Edmondson–Steiner classification (1954), which ranges from Grade 1 to 4, with higher grades showing greater atypia. This histological classification correlates with genetic mutations and cellular proliferation rates [5]. According to the WHO (2019), approximately 65% of HCC cases are categorized as not otherwise specified (NOS-HCC), while the remainder fall into subtypes such as steatohepatitic, clear cell, macrotrabecular-massive, and fibrolamellar [6]. Major risk factors for HCC include chronic hepatitis B and C, non-alcoholic steatohepatitis (NASH), alcohol-related cirrhosis, and hemochromatosis [7]. Early detection is essential, especially in cirrhotic patients. Treatment options range from curative resection and liver transplantation to locoregional therapies such as TACE (Transarterial Chemoembolization), TARE (Transarterial Radioembolization), RFA (Radiofrequency Ablation), and MWA (Microwave Ablation) [8].

Cholangiocarcinoma (CC) is a malignant tumor originating from the epithelial cells of the bile ducts. Based on anatomical location, it is classified into intrahepatic (ICC or Klatskin tumor), perihilar (PCC), and distal (DCC) subtypes. According to the WHO classification, ICC is further divided into two histological subtypes: the large duct type, which resembles extrahepatic cholangiocarcinoma, and the small duct type, which shares imaging and pathological features with HCC [9,10]. While HCC is frequently associated with cirrhosis, ICC is more often linked to risk factors such as primary biliary cholangitis, viral hepatitis (HBV and HCV), metabolic disorders (obesity, diabetes), and liver cirrhosis, although the latter is less prevalent in ICC than in HCC. Other risk factors include biliary cysts, hepatolithiasis, nonalcoholic fatty liver disease, and, more rarely, hemochromatosis and chronic pancreatitis [11]. Surgical resection with negative margins (R0) remains the gold standard for curative treatment of ICC, often combined with lymphadenectomy for accurate staging. Liver transplantation is generally not recommended due to high recurrence risk. In unresectable disease, palliative treatments such as TACE, TARE, and systemic chemotherapy (typically gemcitabine and cisplatin) are used. Advances in molecular profiling have enabled targeted therapies for FGFR2 fusions and IDH1 mutations, supporting personalized treatment strategies [12].

Accurate and timely differentiation between the two most common tumor types is essential for determining appropriate treatment strategies and prognostic evaluation. Despite advancements in imaging technologies, distinguishing HCC from ICC remains a clinical challenge due to overlapping morphological and enhancement features on conventional imaging modalities such as computed tomography (CT) and magnetic resonance imaging (MRI) [13,14,15,16].

Diffusion-weighted imaging (DWI), a functional MRI technique that measures the diffusion of water molecules within tissues, has gained attention as a valuable tool in oncologic imaging [17,18,19]. The apparent diffusion coefficient (ADC), derived from DWI, offers quantitative information about tissue cellularity and structural integrity. Several studies suggest that ADC values differ significantly between malignant and benign lesions and may also differ among tumor subtypes. Specifically, HCC and ICC may exhibit distinct diffusion characteristics owing to their histopathological differences [20,21,22,23].

However, the diagnostic utility of ADC values in distinguishing between HCC and ICC remains insufficiently validated. Reported thresholds and accuracies vary across studies, and limited data are available on the performance of ADC in a histologically confirmed cohort.

The aim of this study is to evaluate the ability of ADC values obtained from diffusion-weighted MRI to differentiate between HCC and ICC, using histological confirmation as the gold standard. By identifying specific ADC thresholds with high diagnostic performance, this research seeks to support the integration of DWI into noninvasive liver cancer protocols.

## 2. Materials and Methods

This retrospective observational study was conducted at Riga East University Hospital. The study was approved by the local ethics committee, and all procedures involving human participants were in accordance with ethical standards of the institutional research committee and with the 1964 Helsinki Declaration and its later amendments [24].

The study included patients who underwent liver MRI and biopsy at Riga East University Hospital between 2019 and 2024. Inclusion criteria consisted of patients with liver lesions who underwent percutaneous liver biopsy resulting in histological confirmation of either hepatocellular carcinoma (HCC) or intrahepatic cholangiocarcinoma (ICC). Patients were excluded if imaging data were incomplete, histological confirmation was unavailable, or ICC maps could not be generated due to technical limitations.

MRI examinations were performed using a 1.5 Tesla scanner equipped with a phased-array body coil. The standard liver MRI protocol included axial T1- and T2-weighted sequences, dynamic contrast-enhanced imaging, and diffusion-weighted imaging (DWI) with b-values of 0, 500, and 1000 s/mm^2^. This b-value scheme reflects the institutional standard protocol, optimized for abdominal imaging to balance sensitivity to diffusion with acceptable signal-to-noise ratio and scan time. ADC maps were automatically generated and calculated from recent MRI scans. For earlier examinations where ADC maps were not included in the imaging protocol, maps were manually generated using a GE HealthCare imaging workstation (GE HealthCare, Chicago, IL, USA).

Apparent diffusion coefficient (ADC) values were calculated from DWI sequences. Regions of interest (ROIs) were manually drawn over the solid portion of the lesion, avoiding necrotic, hemorrhagic, or cystic areas based on co-registered T2-weighted and ADC images. ROIs were placed on the axial slice demonstrating the largest lesion diameter. Mean ADC values (×10^−3^ mm^2^/s) and standard deviations within the ROI were recorded. All measurements were performed by a board-certified radiologist with 10 years of experience in abdominal MRI, and interobserver reproducibility was assessed by a second radiologist with 15 years of experience.

Statistical analysis was performed using IBM SPSS Statistics version 29.0 (IBM Corp., Armonk, NY, USA) [25]. Descriptive statistics were used to summarize patient characteristics and ADC values, expressed as means ± standard deviations. Differences in mean ADC values between HCC and ICC groups were assessed using the independent samples *t*-test. Effect size was calculated using Cohen’s d to evaluate the magnitude of difference. For subgroup analysis by tumor differentiation grade, one-way ANOVA was applied. To evaluate the diagnostic performance of ADC values in differentiating HCC from ICC, receiver operating characteristic (ROC) curve analysis was conducted. The area under the curve (AUC) was calculated to assess overall diagnostic accuracy. An optimal ADC cutoff value was determined based on the Youden index, and corresponding sensitivity, specificity, positive predictive value (PPV), and negative predictive value (NPV) were reported. A *p*-value of <0.05 was considered statistically significant for all comparisons.

## 3. Results

### 3.1. Baseline Characteristics

A total of 736 patient records with suspected or confirmed liver malignancy were reviewed from the archives of Riga East University Hospital. Among these, 323 patients underwent percutaneous liver biopsy. Histological confirmation identified various liver tumors or metastases, but only cases of hepatocellular carcinoma (HCC) or intrahepatic cholangiocarcinoma (ICC) were considered for inclusion. After excluding patients without available MRI scans or without immunohistochemical (IHC) confirmation, 61 patients were included in the final analysis. The final study cohort consisted of 41 patients with HCC and 20 patients with ICC. The mean age was 63.5 years in the HCC group and 65.4 years in the ICC group. The gender distribution was 25 males and 16 females in the HCC group, while the ICC group included 5 males and 15 females. Clinical background factors differed between groups. Viral hepatitis was present in 29.3% of HCC patients and in 10% of ICC patients. Liver cirrhosis was noted in 17.1% of HCC cases and 10% of ICC cases. The clinicopathological characteristics of the patients are presented in Table 1.

### 3.2. Relationship Between ADC Values and Tumor Differentiation

The mean ADC value reflects the average level of water molecule diffusion within the tissue, measured in the selected region of interest (ROI). It is typically expressed in units of ×10^−3^ mm^2^/s. The deviation, or standard deviation, indicates how much the ADC values vary from the mean within the ROI. It reflects the heterogeneity of diffusion, which in turn can suggest variability in tissue structure. A higher deviation may indicate the presence of necrosis or a more aggressive tumor phenotype.

The differentiation grades of HCC and ICC ranged from Grade 1 to Grade 3; Grade 4 was not observed in any tumor type. Out of all the samples analyzed, two were not assigned a differentiation grade and were excluded from the study. The distribution of the remaining biopsies was as follows: Grade 1–8 HCC and 2 ICC; Grade 2–25 HCC and 16 ICC; Grade 3–8 HCC and 2 ICC.

The mean ADC values for well-differentiated, moderately differentiated, and poorly differentiated HCC were 1.23 ± 0.20, 1.10 ± 0.16, and 0.94 ± 0.16 × 10^−3^ mm^2^/s (ranges: 0.99–1.53; 0.76–1.39; 0.69–1.17). Mean ADC in all HCC cases was 1.09 × 10^−3^ mm^2^/s, deviation 164.02 × 10^−3^ mm^2^/s. The ADC values obtained in this study are illustrated in Figure 1, showing representative cases.

In ICC, the calculated ADC values for well-, moderately, and poorly differentiated tumors were 0.98 ± 0.21, 1.09 ± 0.08, and 1.12 ± 0.28 × 10^−3^ mm^2^/s (ranges: 0.83–1.13; 0.96–1.23, 0.92–1.32). Mean ADC in all ICC cases was 1.08 × 10^−3^ mm^2^/s, deviation 161.90 × 10^−3^ mm^2^/s.

### 3.3. Tumor Differentiation and Immunohistochemical Markers

Histopathological analysis was performed on all biopsy samples to determine the tumor differentiation grade and confirm the histological subtype. Tumor differentiation was graded on a three-tier scale (Grade 1–3), with Grade 1 representing well-differentiated tumors and Grade 3 representing poorly differentiated lesions. No cases with grade 4 differentiation were identified.

Immunohistochemistry (IHC) was used to support histological classification, particularly in challenging cases. In cases where morphological features were inconclusive, the IHC profile was essential in distinguishing HCC from ICC, particularly in poorly differentiated tumors.

This study demonstrates that specific immunohistochemical (IHC) markers can effectively differentiate between hepatocellular carcinoma and intrahepatic cholangiocarcinoma. To evaluate the diagnostic relevance of IHC markers in HCC, Fisher’s exact test was applied to compare their expression with ICC. CK-7 (*p* < 0.001), TTF-1 (*p* = 0.0104), and Glypican 3 (*p* = 0.0140) showed statistically significant differences in expression between HCC and ICC, indicating strong discriminatory value. Notably, CK-7 was highly specific for ICC, while TTF-1 and Glypican 3 were more frequently expressed in HCC (Table 1). CK-19 (*p* = 0.0601) and CD34 (*p* = 0.0714) approached significance, suggesting potential diagnostic utility with larger sample sizes. In contrast, CK-20 (*p* = 0.6059) and AFP (*p* = 1.0000), showed no significant difference, limiting their value in distinguishing HCC from ICC in this cohort. These findings support the inclusion of markers such as Glypican and TTF-1 in diagnostic panels targeting HCC, while cautioning against reliance on non-specific or variably expressed markers.

The overall rate of missing IHC data varied considerably across markers. Several markers were not performed in most or all cases, including napsin, chromogranin, and synaptophysin, precluding statistical analysis. No statistical comparisons were conducted for these markers due to complete or near-complete data absence. Nevertheless, their expression patterns—where available—suggest potential diagnostic relevance, warranting further evaluation in larger cohorts. These limitations underscore the importance of consistent and comprehensive IHC profiling in future studies.

In addition, the Ki-67 proliferation index was assessed to evaluate tumor cell proliferation. The average Ki-67 index was 48.57% in HCC (*n* = 7) and 40.75% in ICC (*n* = 4), suggesting relatively high proliferative activity in both tumor types.

### 3.4. Imaging Features of HCC and ICC

A detailed radiological assessment was performed for each histologically conformed liver lesion based on pre-biopsy MRI scans. The following imaging characteristics were evaluated: lesion shape, size, signal intensity on T1- and T2-weighted sequences, arterial phase hyperenhancement (APHE) patterns, presence of capsule, contrast enhancement dynamics, appearance on diffusion-weighted imaging (DWI), hepatobiliary phase findings, and additional features such as necrosis, bile duct dilation, and vascular invasion.

On T1-weighted sequences, most lesions were hypointense relative to surrounding liver parenchyma. T2-weighted images typically showed mild to marked hyperintensity and was more prominent in ICC cases. 

Non-rim arterial phase enhancement (non-rim APHE), a hallmark of HCC, was commonly observed in HCC lesions (Figure 2B). In contrast, rim-APHE and progressive centripetal enhancement were significantly more prevalent in ICC (Figure 3B–D). Delayed centripetal enhancement was present in the ICC group only (Figure 3D). The presence of a capsule was characteristic of HCC. Figure 2 shows a dynamic liver MRI examination using a gadolinium-based contrast agent demonstrating a typical enhancement pattern of an HCC lesion.

Targetoid appearance on DWI was observed in ICC lesions exclusively (Figure 3F), while HCC often presented with more homogenous diffusion restriction. A similar trend was noted on hepatobiliary phase (HBP) imaging, where targetoid features and bile duct dilation were highly specific for ICC. Hepatobiliary phase hypointensity was seen in nearly all lesions, with HCC showing a more varied pattern depending on differentiation and liver function. Well-differentiated tumors were more likely to appear iso- or slightly hypointense in the hepatobiliary phase relative to surrounding liver parenchyma [26]. This is likely due to partial preservation of organic anion transporting polypeptide (OATP) transporter expression, allowing residual hepatocyte function and uptake of hepatospecific contrast agents such as gadoxetic acid [26,27].

The lesion shape was classified as regular or irregular. The majority of HCC lesions exhibited a regular contour, whereas ICC lesions more frequently showed irregular margins, consistent with previously published literature [28,29]. Tumor size ranged from 1.2 to 16.3 cm (mean: 6.62 ± 3.86 cm) in the HCC group and from 1.9 to 14.5 cm (mean: 7.82 ± 4.29 cm) in the ICC group.

Based on LI-RADS (Liver Imaging Reporting and Data System) major imaging features, the HCC lesions corresponded to LI-RADS categories 3 to 5, indicating high probability of correct imaging-based diagnosis [30]. In six HCC cases, features warranted classification as LI-RADS-TIV due to evidence of tumor thrombus in the hepatic veins, suggestive of vascular invasion. This is consistent with findings from Lee et al., who reported that most LI-RADS-TIV observations were malignant, with a high percentage of HCC [31]. By comparison, all ICC lesions were classified as either LI-RADS-3 or LI-RADS-M, reflecting their intermediate or atypical imaging characteristics and limited specificity within the LI-RADS framework [32].

### 3.5. Statistical Analysis

Receiver operating characteristic analysis demonstrated poor diagnostic performance of ADC values in differentiating between HCC and ICC (Figure 4). The area under the curve (AUC) was 0.520 (95% CI: 0.373–0.666, *p* = 0.806), indicating no statistically significant discriminative ability. No optimal threshold was identified; the highest Youden index observed was 0.032 at a cutoff of 1.020 × 10^−3^ mm^2^/s, yielding a sensitivity of 73.2% and a specificity of 30%; and the positive predictive value (PPV) was 68.2%, and the negative predictive value (NPV) was 35.3%.

An independent samples *t*-test showed no statistically significant difference in mean ADC values between HCC (1.09 ± 0.19 × 10^−3^ mm^2^/s) and ICC (1.08 ± 0.11 × 10^−3^ mm^2^/s); *t* (59) = 0.267, *p* = 0.791. The observed mean difference of 0.0122 was small and not clinically relevant, with a negligible effect size (Cohen’s *d* = 0.17). 

A one-way ANOVA revealed a statistically significant difference in mean ADC values across tumor differentiation grades in HCC patients (*F* (2, 38) = 5.50, *p* = 0.008). Mean ADC values decreased with lower differentiation: Grade 1 (1.23 ± 0.20 × 10^−3^ mm^2^/s), Grade 2 (1.10 ± 0.16 × 10^−3^ mm^2^/s), and Grade 3 (0.94 ± 0.16 × 10^−3^ mm^2^/s). Levene’s test indicated homogeneity of variance (*p* = 0.573). The effect size was moderate to large (η^2^ = 0.224), suggesting that differentiation grade accounts for 22.4% of ADC variance. 

In the ICC group, a one-way ANOVA found no statistically significant difference in mean ADC values between differentiation grades (*F*(2, 17) = 0.940, *p* = 0.410). Levene’s test was significant (*p* < 0.001), indicating unequal variances across groups. Mean ADC values were 0.98 ± 0.21 × 10^−3^ mm^2^/s for Grade 1, 1.08 ± 0.08 × 10^−3^ mm^2^/s for Grade 2, and 1.12 ± 0.28 × 10^−3^ mm^2^/s for Grade 3. Due to the small sample size in Grades 1 and 3 (*n* = 2 each), these results should be interpreted with caution, as the limited group sizes may violate ANOVA assumptions and reduce statistical power. To address the limitation of small subgroup sizes, a non-parametric Kruskal–Wallis test was conducted. The analysis confirmed the absence of statistically significant differences in ADC values across ICC differentiation grades (*p* = 0.984). The effect size remained small (η^2^ = 0.100), indicating that only 10% of the variance in ADC values was explained by tumor differentiation grade.

To assess the potential influence of tumor size on ADC values, lesions were classified into two groups based on size: less than 5 cm and 5 cm or larger. No statistically significant differences in mean ADC values were observed between the size-based subgroups in either the HCC (*p* = 0.408) or the ICC (*p* = 0.988) cohort accordingly. Detailed results are presented in Table 2.

## 4. Discussion

Accurately differentiating between HCC and ICC remains a diagnostic challenge, especially in cases with overlapping imaging features. In this study, mean ADC values in HCC (1.09 ± 0.19 × 10^−3^ mm^2^/s) and ICC (1.08 ± 0.11 × 10^−3^ mm^2^/s) were not significantly different (*p* = 0.791), with a negligible effect size (Cohen’s *d* = 0.17), and ROC curve analysis yielded an AUC of 0.520, indicating limited discriminative power. These findings are consistent with prior literature reporting that ADC overlap limits its standalone diagnostic value for distinguishing between these tumor types [20,22,23]. Although an ADC cutoff of 1.020 × 10^−3^ mm^2^/s showed statistical significance, its poor specificity (30%) limits clinical reliability, emphasizing that ADC should not be used in isolation but rather in combination with contrast enhancement patterns, lesion morphology, and clinical context. Notably, we found a statistically significant association between ADC values and tumor grade in HCC (*p* = 0.008, η^2^ = 0.224), with values decreasing as differentiation decreased. This trend reflects the increased cellularity and structural disorganization of poorly differentiated tumors, consistent with earlier reports identifying ADC as a potential surrogate marker for histologic grade in HCC [20,33,34,35]. However, no significant trend was observed across differentiation grades in ICC (*p* = 0.410, η^2^ = 0.100), likely due to small subgroup sizes and tumor heterogeneity. The inconsistent ADC behavior in ICC underscores the need for adjunct diagnostic tools [36].

Our findings highlight distinct MRI features that help differentiate HCC from ICC. Non-rim APHE was commonly observed and aligns with LI-RADS-defined characteristics [13,30,37]. In contrast, ICC lesions more frequently demonstrated rim-APHE, progressive centripetal enhancement, and absence of a capsule—features consistent with prior literature on mass-forming cholangiocarcinoma [13,15,38]. Targetoid diffusion restriction and hepatobiliary phase hypointensity were typical in ICC, while well-differentiated HCCs occasionally retained mild HBP uptake, likely due to preserved OATP transporter expression [27,39]. While radiological patterns support differential diagnosis, they should be interpreted alongside histopathological and immunohistochemical findings to ensure diagnostic accuracy.

Interpretation of ADC values in ICC is further complicated by lesion heterogeneity and the influence of the T2 shine-through effect. In tumors with high T2 signal intensity—often due to necrosis, mucinous content, or fibrosis—hyperintensity on DWI may occur even without true diffusion restriction [40,41,42]. This results in artificially elevated ADC values and can compromise measurement accuracy. The effect is particularly relevant in ICC, which more commonly exhibits such heterogenous components compared to the typically more cellular and homogenous HCC. In this study, several ICC cases were excluded from ADC-based analysis due to poor-quality maps or imaging artifacts consistent with T2 shine-through, highlighting a technical limitation in applying DWI to this tumor type. These findings emphasize the need for careful correlation between ADC measurements and anatomical sequences and suggest that additional diffusion models such as IVIM (Intravoxel Incoherent Motion) or DKI (Diffusion Kurtosis Imaging) may improve lesion characterization in future research [20,23,43,44,45].

Apparent diffusion coefficient (ADC) mapping is supported for use in hepatocellular carcinoma (HCC) for tumor grading, biopsy targeting, and surveillance. Multiple studies demonstrate that lower ADC values are associated with higher tumor grade and increased invasiveness in HCC, making ADC a useful noninvasive biomarker for preoperative tumor grading and risk stratification [35,46]. ADC mapping can also aid in distinguishing well-differentiated HCC from dysplastic nodules, which is relevant for surveillance and early detection [47]. For biopsy targeting, spatial correlation between ADC maps and histopathology has been shown, supporting the use of ADC to guide sampling of the most aggressive tumor regions [48]. For HCC surveillance, non-contrast MRI protocols incorporating diffusion-weighted imaging (DWI) and ADC mapping have demonstrated higher sensitivity and specificity than ultrasound in high-risk populations, supporting their use as an alternative surveillance modality when ultrasound is inadequate [49]. In intrahepatic cholangiocarcinoma (ICC), the utility of ADC mapping is limited by the influence of stromal fibrosis, necrosis, and mucin content on ADC values. While ADC can correlate with tumor grade and stroma in ICC, its specificity is reduced because ADC is affected by non-cellular components such as abundant fibrosis and mucin, which can increase ADC values independent of tumor cellularity [21,50].

Compared to typical hepatocellular carcinoma cohorts, in which cirrhosis is present in up to 80% of cases [51,52], our study reported a substantially lower cirrhosis rate of 17.1% among HCC patients. This discrepancy may reflect differences in referral patterns, biopsy indications, or imaging-based inclusion criteria. Importantly, cirrhosis can influence hepatic parenchymal diffusion characteristics, potentially altering tumor-to-liver contrast on ADC maps. Thus, the low prevalence of cirrhosis in our HCC cohort may limit the generalizability of our findings to broader populations, particularly those with advanced chronic liver disease. Future studies in cirrhotic-dominant cohorts are warranted to validate the diagnostic and prognostic utility of ADC measurements under typical clinical conditions.

The differential diagnosis between HCC and ICC remains a histopathological challenge as well, particularly in poorly differentiated tumors or small biopsies. In this study, several IHC markers demonstrated significant potential for distinguishing HCC from ICC. Glypican 3 and TTF-1 were both significantly more expressed in HCC (*p* = 0.0140 and *p* = 0.0104, respectively), highlighting their value in confirming hepatocellular origin [28,53]. CD34, while not reaching statistical significance (*p* = 0.0714), showed a strong trend toward HCC-specific expression and remains diagnostically useful, especially in combination with other markers [54,55]. Conversely, CK-7, which was 100% positive in ICC and only 30% in HCC, showed highly significant differential expression (*p* < 0.001), supporting its established role as a key marker for cholangiocytic differentiation [56,57,58]. CK-19 also favored ICC (85,7% vs. 33.3% in HCC), with borderline significance (*p* = 0.0601), suggesting its complementary role in panels rather than as a standalone discriminator [59,60,61]. Markers like CK-20 and AFP did not show significant differential expression (*p* > 0.6), limiting their diagnostic specificity in this context. Additionally, neuroendocrine markers were uniformly negative, consistent with the non-neuroendocrine nature of these tumor types. The Ki-67 index may offer supplementary insight into tumor aggressiveness but requires further validation in larger samples [62,63].

Overall, while ADC may assist in tumor grading, it lacks sufficient discriminatory capacity to replace biopsy. Future studies should explore integrated diagnostic models combining imaging, IHC, and emerging molecular markers.

## 5. Conclusions

This study evaluated the diagnostic potential of ADC values in differentiating HCC from ICC based on histologically confirmed cases. While ADC values decreased significantly with lower tumor differentiation in HCC (*p* = 0.008, η^2^ = 0.224), no consistent pattern or statistically significant difference was observed across ICC grades (*p* = 0.410, η^2^ = 0.100).

Overall, mean ADC values did not differ significantly between HCC and ICC, and ROC curve analysis demonstrated poor diagnostic performance (AUC = 0.520, *p* = 0.806). These findings indicate that ADC values alone have limited value in reliably distinguishing between the two mentioned tumor types.

Conversely, immunohistochemical markers such as CK-7, Glypican 3, and TTF-1 showed significant discriminatory potential and may support histological differentiation.

Ultimately, biopsy with histological and IHC evaluation remains mandatory for definitive diagnosis, particularly in poorly differentiated or morphologically ambiguous cases. Future studies with larger cohorts are needed to refine non-invasive diagnostic strategies through integrated imaging–pathology frameworks.

## Figures and Tables

**Figure 1 diagnostics-15-01861-f001:**
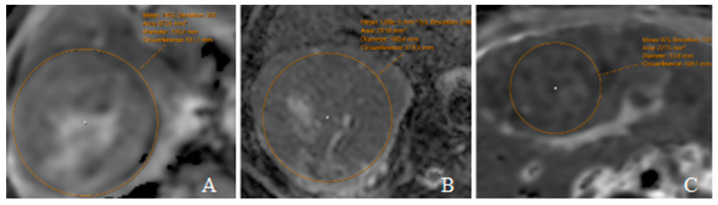
ADC maps in HCC patients with different tumor differentiation grades. (**A**) Grade 1 HCC in a 76-year-old male with a mean ADC value of 1.46 × 10^−3^ mm^2^/s and a deviation of 305 × 10^−3^ mm^2^/s; (**B**) Grade 2 HCC in a 68-year-old male—ADC 1.08 × 10^−3^ mm^2^/s, deviation 294 × 10^−3^ mm^2^/s; (**C**) Grade 3 HCC in a 46-year-old female—ADC 0.98 × 10^−3^ mm^2^/s, deviation 102 × 10^−3^ mm^2^/s.

**Figure 2 diagnostics-15-01861-f002:**
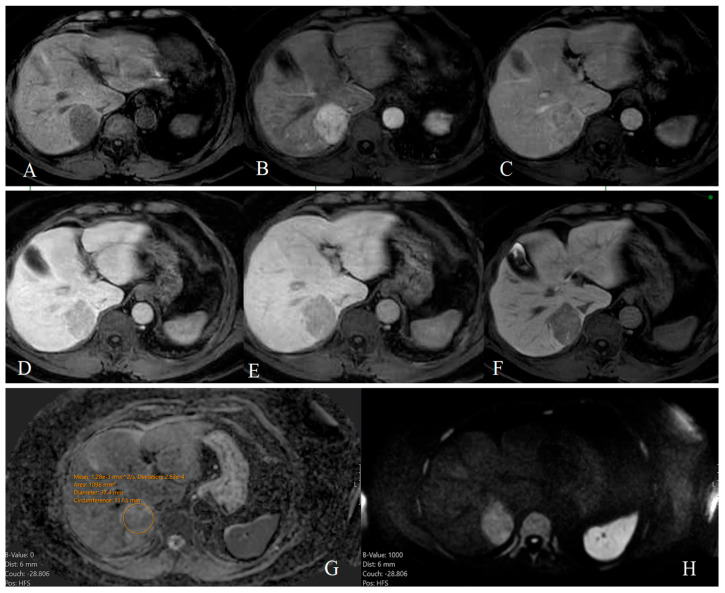
Dynamic MRI with Gd-EOB-DTPA (Primovist^®^) in an 80-year-old male with histologically confirmed moderately differentiated (Grade 2) hepatocellular carcinoma. In the native T1-weighted series, a hypointense lesion is visualized in segment VII of the liver (**A**). The lesion demonstrates non-rim arterial phase hyperenhancement (APHE), (**B**), followed by non-peripheral washout (**C**–**E**) and capsule appearance in the portal venous phase (**D**). It appears hypointense in the hepatobiliary phase (HBP), (**F**) 20 min after contrast administration. On diffusion-weighted imaging (DWI), (**H**), the lesion shows moderate diffusion restriction compared to the spleen. The corresponding ADC value is 1.28 × 10^−3^ mm^2^/s (b = 1000 mm^2^/s) with a standard deviation of 263 × 10^−3^ mm^2^/s (**G**).

**Figure 3 diagnostics-15-01861-f003:**
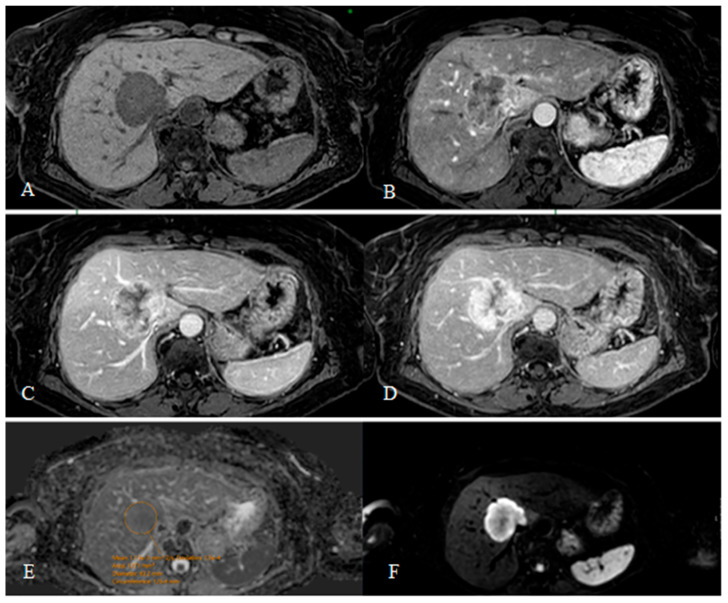
Dynamic MRI in a 65-year-old female with histologically confirmed well-differentiated (Grade 1) intrahepatic cholangiocarcinoma. The lesion appears markedly hypointense on T1-weighted images (**A**). After intravenous bolus administration of the contrast agent, intense peripheral arterial phase enhancement is observed (rim APHE, **B**). In the portal venous phase, the lesion shows inhomogeneous enhancement with progressive centripetal filling (**C**). At 5 min post-contrast (delayed phase, **D**), enhancement increases further, with non-enhancing areas likely corresponding to necrosis. On diffusion-weighted imaging (DWI, **F**), the lesion shows targetoid diffusion restriction. The ADC value is 1.13 × 10^−3^ mm^2^/s (b = 1000 mm^2^/s), with a standard deviation of 150 × 10^−3^ mm^2^/s (**E**).

**Figure 4 diagnostics-15-01861-f004:**
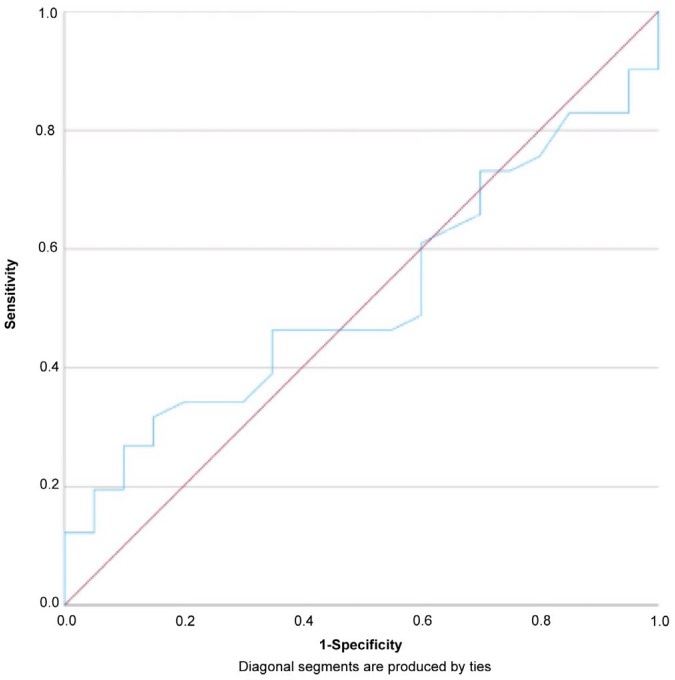
ROC curve for ADC values in differentiating HCC from ICC.

**Table 1 diagnostics-15-01861-t001:** Clinical features of patients with hepatocellular carcinoma and intrahepatic cholangiocarcinoma.

Variables	HCC (*n* = 41)	ICC (*n* = 20)	*p*-Value
Sex, *n* (%)			0.013
Male	25 (60.98%)	5 (25%)	
Female	16 (39.02%)	15 (75%)	
Age (years)			0.572
Max	86	90	
Min	29	36	
Mean	63.48 ± 12.65	65.35 ± 11.67	
Viral hepatitis, *n* (%)	12 (29.27%)	2 (10%)	0.116
Radiographic examination			
Liver cirrhosis, *n* (%)	7 (17.07%)	2 (10%)	0.704
Size (max cm), *n* (%)			0.453
≤5	16 (39%)	6 (30%)	
>5 and ≤10	15 (37%)	6 (30%)	
>10	10 (24%)	8 (40%)	
Pathological examination			
Differentiation, *n* (%)			0.332
Grade 1	8 (20%)	2 (10%)	
Grade 2	25 (61%)	16 (80%)	
Grade 3	8 (20%)	2 (10%)	
Necrosis, *n* (%)	5 (12%)	3 (15%)	1.000
Immunohistochemistry, P/N samples (%)			
TTF-1	15/7 (68.2%)	2/10 (16.7%)	0.010
CD-34	15/4 (78.9%)	0/2 (0.0%)	0.071
CK-7	6/14 (30.0%)	16/0 (100.0%)	<0.001
CK-19	3/6 (33.3%)	6/1 (85.7%)	0.060
CK-20	1/12 (7.7%)	3/13 (18.8%)	0.606
CK-AE-1/AE-3	8/0 (100%)	3/0 (100.0%)	NA *
Glypican 3	16/9 (64.0%)	0/3 (0.0%)	0.067
AFP	2/7 (22.2%)	0/3 (0.0%)	1.000
p53	1/1 (50.0%)	1/1 (50.0%)	1.000
Napsin	0/1 (0.0%)	–	NA
Chromogranin	0/6 (0.0%)	0/1 (0.0%)	NA
CD-10	2/2 (50.0%)	–	NA
Synaptophysin	0/6 (0.0%)	0/4 (0.0%)	NA
Reticulin (reduced or lost)	4/9 (30.8%)	–	NA
Ki-67 (%)	48.57 (7 samples)	40.75 (4 samples)	0.700

HCC, hepatocellular carcinoma; ICC, intrahepatic cholangiocarcinoma; Max, maximum; Min, minimum; TTF, Thyroid Transcription Factor; CD, Cluster of Differentiation; CK, Cytokeratin; AFP, Alpha-Fetoprotein; P, positive; N, negative. * NA: *p*-values were not calculated for markers where all samples were positive or negative, or where the test was statistically invalid due to lack of variance.

**Table 2 diagnostics-15-01861-t002:** ADC values by tumor size in HCC and ICC.

Type	Size Group	*n*	Grade 1	Grade 2	Grade 3	Mean ADC (×10^−3^ mm^2^/s)	SD ADC	*p*-Value
HCC	<5 cm	14	2	10	2	1.06	0.18	0.4075
	≥5 cm	27	6	15	6	1.11	0.19	0.4075
ICC	<5 cm	5	0	5	0	1.08	0.07	0.9883
	≥5 cm	15	2	11	2	1.08	0.13	0.9883

HCC, hepatocellular carcinoma; ICC, intrahepatic cholangiocarcinoma; ADC, apparent diffusion coefficient; SD, standard deviation.

## Data Availability

The raw data supporting the conclusions of this article will be made available by the authors on request.

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
