# Peer review of "Comparison of Apparent Diffusion Coefficient Values on Diffusion-Weighted MRI for Differentiating Hepatocellular Carcinoma and Intrahepatic Cholangiocarcinoma"

_diagnostics, 2025, doi:10.3390/diagnostics15151861_

Round 1

Reviewer 1 Report

Comments and Suggestions for Authors

Manuscript title: "Comparison of Apparent Diffusion Coefficient Values on Diffusion-Weighted MRI for Differentiating Hepatocellular Carcinoma and Intrahepatic Cholangiocarcinoma"

1) The study aims to evaluate the diagnostic utility of apparent diffusion coefficient (ADC) values derived from diffusion-weighted MRI (DWI) in distinguishing hepatocellular carcinoma (HCC) from intrahepatic cholangiocarcinoma (ICC) with histopathological confirmation as the reference standard. Secondary objectives include assessing relationship with tumor differentiation grades and correlating findings with immunohistochemical (IHC) markers.

2) Distinguishing HCC and ICC noninvasively is valuable for treatment planning, but remains challenging due to overlapping imaging features. This work adds to the available body of evidence by rigorously testing ADC's real-world applicability. The study also contributes robust histology-confirmed data to debates about ADC's discriminative capacity. Additionally, validation of IHC markers (CK-7, Glypican 3, TTF-1) reinforces their role in ambiguous cases.

3) The conclusions align with the evidence provided:

  • Lack of significant ADC difference between HCC and ICC (p=0.791, AUC=0.520) is supported by statistical analyses and is consistent with prior literature on ADC overlap.

  • Association between lower ADC and poorer HCC differentiation is well-substantiated (p=0.008, η²=0.224).

  • Emphasis on IHC for definitive diagnosis (especially CK-7 for ICC, Glypican 3/TTF-1 for HCC) is justified by significant p-values (<0.05).

4) The reviewer's suggestions to improve the study are as follows:

  • Clarify how ROIs avoided necrosis/hemorrhage (e.g., reference to co-registered T2/contrast sequences). Specify if ROIs were placed on single or multiple slices and operator experience level to ensure reproducibility.

  • Report echo/repetition times, coil type, and respiratory compensation methods. Justify b-value selection (0,500,1000 s/mm2) versus multi-b-value models (IVIM?), which may better address confounding factors like perfusion.

  • Explicitly discuss ICC subgroup limitations (Grades 1/3, n=2) and their impact on ANOVA validity. Consider sensitivity analyses or non-parametric tests.

  • Expand Fisher’s exact test to include adjusted p-values (Benjamini-Hochberg) for multiple comparisons, reducing false discovery risk. Report missing data rates for IHC markers. Clearly state that this test will be used in Materials and Methods.
  • Clarify that the identified cutoff (1.020 ×10⁻³ mm2/s) has poor specificity (30%) and may be clinically unreliable.

  • Discuss how ADC’s role in HCC grading could influence surveillance (e.g., monitoring dedifferentiation) or biopsy targeting. Contrast this with its limitations for ICC.

  • Compare cohort demographics (e.g., lower cirrhosis rates in HCC [17.1%] vs. typical cohorts [~80%]), which may affect ADC generalizability.

5. The Tables and Figures are informative.

Reviewer 2 Report

Comments and Suggestions for Authors

This manuscript compared the apparent diffusion coefficient (ADC) values on diffusion-weighted MRI for differentiating hepatocellular carcinoma and intrahepatic cholangiocarcinoma. 

I have a few questions.

1. In Table 1, there were no p values of each comparison. Please provide the statistical significance of the results.

2. I think that tumor size and differentiation could be classified into two groups for enhancing the statistical power for the comparison.  For example, tumor size < or >= 5cm and histologic grade >=3 (poor) and 1or 2.

And then you could perform multivariate analysis for its comparison. 

3. How about other preoperative clinical factors such as AFP, PIVKA, CA 19-9, CEA? These tumor markers also represents the tumor biology predicting poor histologic grade. There factors could Multivariate analysis could be preformed with other clinical factors for predicting poor histologic grade, especially in HCC.

Comments on the Quality of English Language

Well written, but it can be more improved. 

Round 2

Reviewer 1 Report

Comments and Suggestions for Authors

The authors have provided comprehensive revisions and detailed responses. The manuscript has been improved.

  • Comment 1 (ROI methodology): response accepted.
  • Comment 2 (MRI parameters): response accepted.
  • Comment 3 (ICC subgroup limitations): response accepted.
  • Comment 4 (Statistical adjustments): response accepted.
  • Comment 5 (ADC cutoff specificity): response accepted.
  • Comment 6 (ADC clinical role): response accepted.
  • Comment 7 (Cohort generalizability): response accepted.